# Environmental conditions rather than nitrogen availability limit nitrous oxide (N₂O) fluxes from a temperate birch forest

Galina Y. Toteva[1,2], David Reay[1], Matthew Jones[2], Ajinkya Deshpande[2], Nicholas Cowan[2], Peter Levy[2], Duncan Harvey[2], Agata Iwanicka[2], Julia Drewer[2]

[1]School of GeoSciences, The University of Edinburgh, Edinburgh, EH9 3FE, United Kingdom

[2]UK Centre for Ecology & Hydrology, Bush Estate, Penicuik, EH26 0QB, United Kingdom

*Correspondence to*: Galina Toteva (galtot@ceh.ac.uk) and Julia Drewer (juew@ceh.ac.uk)

Key words: ammonia deposition, NH₃, methane, climate change, soil, NO

**Abstract.** Forest ecosystems play an important role in the terrestrial nitrogen (N) cycle, accounting for over a quarter of the land area of the Earth. However, our understanding of nitrogen dynamics in forest systems is limited. The consequences of N deposition to forest ecosystems are often overlooked. In this study, dry deposition of $NH_3$ was replicated over a two-year period in a temperate semi-natural birch forest via a unique custom-built automated $NH_3$ release system to investigate the impact on emissions of the greenhouse gas nitrous oxide ($N_2O$). This study provides evidence that in both natural forest soils (in-situ) and soils under controlled laboratory conditions (ex-situ), the substantial addition of reduced N compounds ($NH_3$/$NH_4^+$) had no direct impact on $N_2O$ emissions. Emissions of $N_2O$ from these soils were dependant on the meeting of several additional thresholds, below which $N_2O$ producing activity was constrained. When environmental conditions in-situ were considered warm and wet (soil temperature >12 °C and volumetric water content >20%), emissions of $N_2O$ were an order of magnitude higher than when either of these thresholds was not met, regardless of exposure to $NH_3$ deposition. Ex-situ experiments indicated that microbial activity in the soils was highly constrained by the availability of labile carbon. The addition of glucose to these soils resulted in a considerable increase in $N_2O$ emissions after N application. While cumulative $NH_3$ deposition to the in-situ soils was relatively large over the measurement period, there was no accumulation of mineral N observed in the soil, suggesting plant-uptake of N was able to mitigate N loading. The implication of these results is that forest ecosystems may be able to mitigate localised $NH_3$ pollution plumes, in the short-term at least, without incurring an $N_2O$ penalty. However, the long-term impacts of N enhancement remain unclear and further long-term field experiments are required to examine the impact of prolonged exposure to high quantities of N deposition to forest soils.

**Short summary.** The impacts of increasing nitrogen deposition on the fluxes of the greenhouse gas nitrous oxide from a temperate birch forest were investigated in-situ and ex-situ. Nitrogen levels only had a limited effect on emissions. Instead, emissions of nitrous oxide were modulated by soil carbon availability and meeting a dual temperature-moisture threshold. An implication of these findings is that forests could be used for mitigating nitrogen pollution without incurring a greenhouse gas penalty, at least in the short term.

## 1. Introduction

Reactive nitrogen ($N_r$) entering ecosystems due to anthropogenic activities has more than tripled at the global scale relative to 1961 (Galloway et al., 2021). Consequently, $N_r$ pollution has contributed significantly to loss of biodiversity (Bobbink et al., 2010; Krupa, 2003), eutrophication (de Vries, 2021), soil acidification (Tian and Niu, 2015) and gaseous emissions in the form of nitrous oxide ($N_2O$) and nitric oxide (NO) (Butterbach-Bahl et al., 2013; Davidson et al., 2000; Pilegaard, 2013; Song et al., 2020). $N_2O$ is a potent greenhouse gas (GHG) with a global warming potential approximately 273 times higher than carbon dioxide ($CO_2$) on a 100-year timescale (IPCC, 2023) and is also responsible for destruction of ozone in the stratosphere (Ravishankara et al., 2009). Natural soils are an important source of $N_2O$, representing approximately 35% (6.4 Tg N $yr^{-1}$) of the global $N_2O$ budget (Tian et al., 2024). The production of both $N_2O$ and NO gases in soils is generated as a by-product of microbial processes, predominantly via nitrification and denitrification (Baggs, 2011; Butterbach-Bahl et al., 2013; Pilegaard, 2013). These processes and the magnitude of resulting emissions from soils are affected by a wide range of environmental variables, such as temperature (Braker et al., 2010), pH (Weslien et al., 2009), soil moisture (Firestone and Davidson, 1989), and carbon (C) and nitrogen (N) availability (Butterbach-Bahl et al., 2013; Pilegaard, 2013; Skiba and Smith, 2000).

Nitrogen availability in soils is considered to be a major driver of $N_2O$ fluxes. While generally considered a rate-limiting step in low-N natural environments, higher availability of $N_r$ would be expected to result in increased microbial activity and higher gaseous losses from soils in the form of $N_2O$ and NO fluxes (Firestone and Davidson, 1989). While this pattern is generally observed following the application of N-based fertiliser to agricultural systems (e.g. Cowan et al., 2020), this is not always the case in natural ecosystems, such as forests. The latter are typically exposed to chronic N inputs, primarily from atmospheric deposition (Sutton et al., 2004; Sutton et al., 2014), and often do not demonstrate a clear dose-response to elevated N levels (Du and de Vries, 2023). While some synthesis studies have attributed an increase in $N_2O$ forest fluxes to elevated N levels (Aronson and Allison, 2012; Cen et al., 2024), others have reported a lack of response and high variability in the fluxes (Flechard et al., 2020; Liu and Greaver, 2009).

In addition to the amount of $N_r$, the form of reduced or oxidised N affects the rates of different microbial processes and their associated $N_2O$ and NO emissions (Ding et al., 2023). In the UK, a shift in policies over the past several decades has significantly decreased emissions of oxidised $N_r$ (e.g. $NO_x$) (Driscoll et al., 2024; Tomlinson et al., 2021) and shifted the ratio

of N deposition further towards reduced forms such as ammonia ($NH_3$) (Hicks et al., 2022; Tomlinson et al., 2021). $NH_3$ is the primary contributor to $N_r$ pollution in the UK (Tang et al., 2018). An estimated 260 kt of $NH_3$ was emitted in the UK in 2022, with agricultural activities and fuel combustion (mainly from transportation) being the main sources (Mitchell et al., 2024). Volatilised $NH_3$ can deposit to natural ecosystems, such as forests, where it acts as a major source of $N_r$ (Bobbink et al., 2010; Du and de Vries, 2023). It is estimated that between 7 to 50 kg N ha$^{-1}$ yr$^{-1}$ are deposited to UK forests, with approximately 90% of forests being subjected to a critical load exceedance (the level of N deposition above which negative impacts occur) (Vanguelova et al., 2024).

Dry deposition of $NH_3$ is an often-overlooked source of $N_r$ to forests (Du and de Vries, 2023; Flechard et al., 2011). It has been estimated that 63% of $N_r$ deposition to European forests is due to dry deposition (Flechard et al., 2020). Dry deposition is likely to play a more important role in forests compared to other natural ecosystems (such as grasslands and moorlands) due to complex canopy structures and multiple surfaces for $N_r$ deposition (Vanguelova et al., 2024). This has been demonstrated by the fact that the concentration of nitrate ($NO_3^-$) and ammonium ($NH_4^+$) in throughfall in the UK was higher in forested areas relative to moorlands and open grasslands (Sawicka et al., 2016).

While a large proportion of studies on N dynamics and the effects of elevated N levels in soils have focused on agricultural systems (Reay et al., 2012), forest ecosystems also play an important role in the terrestrial N cycle (Chapin et al., 2011). Forests cover approximately one third of global land area (Keenan et al., 2015; Ritchie, 2024). This proportion may increase as a result of global (DESA, 2018) and national (Westaway et al., 2023) efforts to increase tree cover as part of climate change mitigation measures. It is believed that increasing tree cover can contribute towards simultaneously mitigating climate change (by sequestering atmospheric $CO_2$) and reducing atmospheric $NH_3$ pollution (Kirschbaum et al., 2024; Tang et al., 2022; Verheyen et al., 2024). However, $NH_3$ deposition to forests can alter natural conditions, such as through changes in tree growth (fertilisation effect), soil C sequestration, understory vegetation diversity and changes in greenhouse gas (GHG) fluxes (such as $CO_2$, methane ($CH_4$) and $N_2O$) (Tang et al., 2022; Vanguelova et al., 2024).

There remains high uncertainty regarding the impacts of a potential increase in soil N availability on $N_2O$ fluxes in forests, and experimental methodology is a major limiting factor when attempting to replicate the complexities of N deposition in forest ecosystems (Du and de Vries, 2023). For example, Jiang et al. (2023) reported that applying N directly to the soil rather than to the forest canopy can increase $N_2O$ fluxes by 20 to 50%, thus highlighting the importance of tree canopies in modulating the impacts of N deposition on soil processes. The form of N (reduced versus oxidised) has also been reported to affect the diversity and abundance of microbial communities (Ding et al., 2023) and as such it might indirectly impact gaseous emissions from the soil. This indicates a research gap and highlights the importance of studying and understanding the impacts of reduced $N_r$ dry deposition on $N_2O$ fluxes from forest soils.

In this study, dry deposition of $NH_3$ was replicated in a temperate semi-natural birch forest via a unique custom-built automated $NH_3$ release system (Deshpande et al., 2024). Anhydrous $NH_3$ gas was released based on wind conditions at low concentrations over two years, thus mimicking a gradient of realistic levels of N deposition across forest soils from which $N_2O$ emissions were measured. A complementary laboratory experiment further investigated the role of other environmental factors, such as N form and soil carbon availability, on $N_2O$ fluxes. The primary aims of this work were to: (i) determine the impact of increasing $NH_3$ dry deposition on $N_2O$ fluxes from forest soils, (ii) identify the environmental factors affecting $N_2O$ fluxes in forest soils and (iii) explore the role of soil carbon availability in modulating $N_2O$ flux in forest soils.

## 2. Methodology

### 2.1. Site description

This study was carried out at Glencorse forest, Midlothian, Scotland, United Kingdom (55°51'13" N, 3°12'56" W; 186 m above sea level) (Fig. S1). It represents a semi-natural temperate forest. The dominant tree species are silver birch (*Betula pendula*) and downy birch (*B. pubescens*) which were planted in 1984 (Billington and Pelham, 1991). Some natural regeneration has occurred, predominantly ash (*Fraxinus excelsior*) and rowan (*Sorbus aucuparia*). The ground vegetation is dominated by grasses (*Calamagrostis stricta*, *Festuca gigantea*, *Holcus lanatus*, *Dactylis glomerata* and *Poa nemoralis*). Previously, the site was used as an agricultural field, similar to the modern use of the fields that surround Glencorse. The site has not been managed and has not received additional nitrogen inputs, other than atmospheric deposition, since the 1990s. Background atmospheric deposition rates of $NH_3$ at the Glencorse field site before the start of the manipulation experiment were approximately 0.63 µg $m^{-3}$ (Deshpande et al., 2024), which falls below the critical N level for the UK (Rowe et al., 2021). Soil type in Glencorse is classified as freely drained brown earth from the Darvel series derived from Carboniferous sediments (Levy and Clark, 2009). Soil physicochemical properties are summarised in Table 1. Mean annual temperature is approximately 9 °C and annual precipitation is approximately 993 mm for the period 1991 to 2020 (UK Met Office, 2023).

**Table 1 Soil physicochemical characteristics at the Glencorse field site (n = 36).**

|  | Mean | SD |
|---|---|---|
| Total carbon, % | 3.40 | 0.79 |
| Total nitrogen, % | 0.26 | 0.05 |
| C:N ratio | 13.1 | - |
| pH | 5.32 | 0.31 |
| Bulk density, g $cm^{-3}$ | 0.96 | 0.15 |

## 2.2. Automated NH₃ release system and atmospheric NH₃ concentrations

A unique custom-built automated $NH_3$ release system was used to increase the atmospheric concentrations of $NH_3$ in the Glencorse study site (Fig. S2) (Deshpande et al., 2024). Anhydrous $NH_3$ gas was mixed with air and blown down three 20-metre-long perforated uPVC pipes (d = 11 cm) at three heights above the ground (0.5, 1.35 and 2.2 m) to facilitate even N enhancement along the soil surface, underground vegetation and trees (Deshpande et al., 2024). Wind speed and wind direction were measured using a weather transmitter (WXT 536, Vaisala, Finland) located 2.3 m above the ground on a meteorological tower. Ammonia was continuously released when the wind was in the South-West sector (275° – 345°) and between 0.3 and 10 m s$^{-1}$. Concentrations of $NH_3$ were highest closest to the release line and decreased downwind with distance away from the source, as observed in the vicinity of chicken farms (Sommer et al., 2009; Pitcairn et.al, 2002), thus creating a gradient of realistic concentrations. The system was activated in September 2021 and has been active since then. The timings and amount of $NH_3$ release were recorded by a Micrologger (CR 3000, Campbell Scientific).

Adapted Low-cost Passive High Absorption ALPHA® samplers (Tang et al., 2001) were used to measure atmospheric $NH_3$ concentrations. These are passive diffusion samplers with a path length of 6 mm. A filter paper coated in 12% citric acid was enclosed by a polyethylene sampler body. A PTFE membrane was placed at the open end of the body, thus allowing for air to diffuse from the atmosphere towards the filter without interference from turbulence. The optimum range of $NH_3$ concentrations that these ALPHA® samplers could measure was between 0.03 and 100 µg m$^{-3}$. Samplers were exposed for one month at a time, thus presenting a cumulative amount of atmospheric $NH_3$ for the specified period. Samplers were positioned within 0.5 m of each static flux chamber (see below) so that $NH_3$ atmospheric concentrations could be correlated to $N_2O$ fluxes without the need for spatial interpolation. Alpha samplers were placed 0.5 m above the ground to capture $NH_3$ concentrations close to the level of the chambers, yet minimising interference from ground vegetation. Following exposure, filters were extracted in 3 ml of deionised water for one hour and analysed using a flow injection analyser based on the salicylate method (Seal AA3 HR AutoAnalyzer, Seal Analytical Ltd., Wrexham, UK).

## 2.3. Measurements of N₂O and CH₄

The in-situ study was designed as a "before-after-control-impact" (BACI) experiment (Christie et al., 2019; Smokorowski and Randall, 2017). A total of 36 static flux chambers were first deployed in March 2021 (six months prior to activation of the $NH_3$ release system) to capture the temporal and spatial variability in $N_2O$ and $CH_4$ fluxes (Fig. S3). Pre-treatment gas fluxes from the soils were measured between March and September 2021. The rest of the flux measurements took place after the activation of the release system. Seven of the 36 chambers were positioned upwind from the $NH_3$ release line to act as a control, where $NH_3$ concentrations were close to background levels. The other 29 chambers were downwind of the release line and hence received elevated atmospheric $NH_3$ deposition to examine the effect of the additional $N_r$.

Gas samples for measuring $N_2O$ concentrations were collected approximately once a month. Metal lids were placed on top of the polyvinyl chloride (PVC) static chambers (inner diameter = 0.38 m, h = 0.12 m on average) to create a headspace of approximately 0.013 $m^3$ during sampling. Draught excluders and bulldog clips were used to ensure airtightness and thus minimise the chances for leakage. Gas samples were collected using a 100-ml syringe via a three-way tap and stored in 20 ml glass vials in line with the double-needle technique described in Drewer et al. (2021). Gas samples were collected every 20 min for a total enclosure time of 60 minutes, resulting in four samples per chamber per sampling event (t0, t20, t40, t60). This is longer compared to some previous studies, in order to ensure that detectable concentrations built up since $N_2O$ fluxes in unfertilised forests are smaller than agricultural systems (Stehfest and Bouwman, 2006). There were no signs of saturation within the chambers.

Gas samples were analysed within one week of collection using a gas chromatograph (GC) (7890B GC system, Agilent Technologies, California, USA). These were interspersed with sets of four standards of known concentrations for quality control, ranging from 208 ppb to 1040 ppb for $N_2O$ and from 1.12 ppm to 98.2 ppm for $CH_4$. The GC system was fitted with a flame ionisation detector (FID) and a micro-electron capture detector (µECD) to measure $CH_4$ and $N_2O$, respectively. The analytical uncertainty in flux methodology was calculated to be $\pm0.05$ nmol $m^{-2}$ $s^{-1}$ for $N_2O$ fluxes and $\pm0.58$ nmol $m^{-2}$ $s^{-1}$ for $CH_4$ fluxes (Cowan et al., 2025). Fluxes of $N_2O$ and $CH_4$ were calculated from the change in concentrations during the enclosure period according to Eq. (1):

$$F = \frac{dC}{dt} \cdot \frac{\rho V}{A} \tag{1}$$

where $F$ is gas flux from the soil (nmol $m^{-2}$ $s^{-1}$), $dC/dt$ is the rate of change in concentration with time in nmol $mol^{-1}$ $s^{-1}$, $\rho$ is the density of air in mol $m^{-3}$, $V$ is the volume of the chamber in $m^3$ and $A$ is the ground area enclosed by the chamber in $m^2$. Calculations were performed using the RCflux package in R (Levy et al., 2011). Linear regression was used to calculate $dC/dt$ for all measurements as it presented the best fit model in the majority of cases, there were no signs of saturation within the chambers and is a commonly used flux calculation method (Levy et al., 2011).

### 2.4. Controlled laboratory experiments

A complementary laboratory experiment was performed in order to test for the effects of N form, N enhancement level, and C availability on $N_2O$ and NO fluxes under controlled conditions. Thirteen soil samples from the top 0-10 cm were collected randomly from the Glencorse forest site in July 2022 from control areas which did not experience elevated $NH_3$ levels and combined to one bulk sample. Soil was dried at 25 °C and sieved through a 2 mm steel sieve. 800 g of dry homogenised soil were placed in Perspex chamber bottoms (d = 19 cm, h = 10 cm) and repacked to field bulk density (0.7 to 1.2 g $cm^{-3}$).

Treatments simulated N deposition equivalent to 0, 40 and 100 kg N ha$^{-1}$ yr$^{-1}$ (hereafter labelled 0N, 40N and 100N, respectively). Target N deposition levels for the laboratory incubations were based on measured values from the in-situ field experiment to allow for continuity and comparability of the results. N was applied in the form of either $NH_4^+$ in aqueous solution (Cowan et al., 2024) or $NH_4NO_3$ in order to study the effects of reduced and oxidised N forms. The application solutions were prepared by adding deionised water to a stock solution to reach a target concentration which simulates N deposition at the target level (0N, 40N or 100N). Consequently, 8.5 or 21.3 mg N were added either as $NH_4^+$ or $NH_4NO_3$ to simulate the 40N and 100N treatments, respectively. Enhanced C availability was achieved through the addition of glucose. A 1% sugar solution was prepared by adding 20 g analytical grade glucose (Sigma Aldrich) to 1 L deionised water (Sanchez-Martín et al., 2008). This is the equivalent of 6.6 g of sugar per soil core (800 g dry soil on average). There were three replicate cores for each treatment (including control) (n = 15), from which $N_2O$ and NO concentrations were measured.

All re-packed soil cores in the incubation chambers were initially saturated with deionised water. The cores were then allowed to gradually dry out to allow for any Birch effect and consequent artifact gaseous emissions to subside (Birch, 1964). Deposition of N was simulated following the method described in Song et al. (2020). First N dose (in the form of either $NH_4^+$ or $NH_4NO_3$) was applied 7 days after the initial rewetting. The second dose (N plus glucose) was applied 34 days after the initial rewetting. $N_2O$ and NO measurements were taken every day during the first week following the treatment application and every second day thereafter for a total of 42 days.

$N_2O$ concentrations were measured using the dynamic chamber method described in Cowan et al. (2014). The experimental setup consisted of a quantum cascade laser (Tildas-FD, Aerodyne Research Inc., Billerica, MA, USA) and a pump used to circulate the air (SH-110, Dry Scroll Vacuum Pump, Agilent Technologies, Lexingtom, MA, USA). A Perspex chamber was attached via bulldog clips to each soil core for the duration of each measurement (on average 5 minutes). The flow rate of air through the system was around 3.5 L min$^{-1}$ on average. The total headspace of the soil cores plus the measuring chamber was 0.01 m$^3$ on average. $N_2O$ concentrations were recorded continuously with frequency of 1 Hz (once every second) for 5 min on average, which resulted in approximately 200 data points per measurement. Concentrations of $N_2O$ were measured as dry mole fraction following an internal water correction within the Aerodyne software. Daily $N_2O$ fluxes were calculated based on the change in concentrations, following the same principle as the in-situ experiment (Eq. (1)). Cumulative $N_2O$ fluxes were calculated using linear interpolation (Cowan et al., 2019).

NO concentrations were measured using a NO-NO$_2$ ultrasensitive chemiluminescence analyser (Model T200UP, Enviro Technology Services plc) as part of a gas flow-through system (Drewer et al., 2015). A pump was used to circulate the air through the system at a flow rate of approximately 1 L min$^{-1}$. Filtered laboratory air was used to make up the volume needed for the analyser. This formed an open loop system, unlike the QCL setup which was a closed loop. The Perspex chambers used for measuring $N_2O$ and NO fluxes were identical (d = 19 cm, h = 20 cm). NO concentrations were measured for 20 to 30

minutes, depending on how quickly the system reached equilibrium. Equilibrium concentrations were recorded for two to four minutes. Concentrations of $O_3$ were monitored (49C $O_3$ analyser, Thermo Environmental Instruments Inc, USA) to ensure that they were below 5 ppb and thus the probability of chemical reactions occurring in the system was low (Seinfeld and Pandis, 220    2016). Temperature and relative humidity inside the chamber were measured using an integrated transmitter (Humitter 50 YC Y 50 10002, Vaisala). These were recorded every 10 seconds using a data logger (CR1000, Campbell Scientific).

Fluxes of NO were calculated using equilibrium concentrations (as opposed to change in concentrations) following the formula outlined in Schindlbacher et al. (2004) (Eq. 2).

$$F = \left(C_{eq} - C_0\right) * \frac{M*Q*10^6}{V_m*A*10^9} * 60 \tag{2}$$

225

Where $F$ is NO-N flux expressed in µg m$^{-2}$ h$^{-1}$, $M$ is atomic weight (N = 14.008 g mol$^{-1}$), $Q$ is the mass flow rate of air through the chamber (1 L min$^{-1}$ on average), $V_m$ is the standard gaseous molar volume (24.055 * 10-3 m$^3$ mol$^{-1}$), $C_{eq}$ is the mixing ratio of NO at equilibrium, $C_0$ is the mixing ratio of NO from an empty chamber (blank), and $A$ is the soil surface area of the soil core (0.03 m$^2$).

**2.5. Soil properties and meteorological conditions**

Inorganic N availability in soils was measured in the form of $NH_4^+$ and $NO_3^-$. During the in-situ experiment, soil samples were collected from the top 10 cm within 0.5 m of each static flux chamber once every season. All samples were frozen within two hours of collection and stored at -20 °C until analysis. 15 g of soil were extracted in 50 ml of 1M potassium chloride (KCl) solution to obtain mineral N. These were mixed at 100 rpm for 60 minutes on an orbital shaker (SSL1 orbital shaker, Stuart) 235    and consequently filtered through Whatman No. 40 ashless filter paper (pore size = 8 µm). The concentrations of KCl-extractable $NO_3^-$ and $NH_4^+$, were measured using a discrete multi-chemistry analyser (Seal AQ 2, Seal Analytical Inc., Wisconsin, USA). Water content was corrected for by measuring and subtracting the gravimetric content of separate samples (dried at 105 °C for three days). A set of blanks (KCl solution only) were analysed alongside all samples to ensure there was no N contamination in the laboratory. Blank values were subtracted from sample values. When sample concentrations were 240    low, this resulted in artefact negative concentrations.

The amount of N in the form of $NH_4$-N and $NO_3$-N was calculated per gram of dry soil according to Eq. (3).

$$N = \frac{c*v}{m} \tag{3}$$

Where $N$ is the mass of N in the form of $NH_4$-N or $NO_3$-N expressed in mg N per g of dry soil; $c$ is the concentration of $NH_4$-N or $NO_3$-N in mg L$^{-1}$; $v$ is the volume of KCl solution used during soil extractions in L; $m$ is the mass of dry soil in g.

Approximately 15 g of soil were collected from the top 10 cm within 0.5 m of each static flux chamber three times throughout the experiment (March 2021, November 2022 and March 2023) to measure total C and N content. Samples were dried at 105 °C for three days (until constant weight) and milled using a ball mill (MM200 ball mill, Retsch). Approximately 2 mg of soil sample were analysed using an elemental analyser (Flash SMART, Thermo Fisher Scientific).

Samples for measuring soil pH were collected and stored on the same dates and in the same way as the samples used for mineral N analysis. pH was measured using a pH meter (MP 200, Mettler Toledo GmbH, Schwerzenbach, Switzerland), where 20 ml of deionised water were added to 10 g of soil sample, shaken and left to rest for 60 minutes prior to measurement. A 2-point calibration using buffer solutions of pH 4 and 7 was performed at the beginning of each measurement day.

Bulk density was measured at the beginning of the experiment (March 2021). Soil was collected from the top 10 cm within 0.5 m of each static flux chamber using a metal ring of known volume (d = 7.5 cm, h = 5 cm). Care was taken to minimise compaction when sampling. Samples were oven-dried at 105 °C for three days or until constant weight was reached. Bulk density was calculated by dividing the dry weight of soil by the soil volume (Robertson et al., 1999).

Soil temperature and soil moisture were recorded in two distinct ways to capture both spatial and temporal variability. Soil temperature and soil moisture were recorded within 0.5 m of each static flux chamber every time gas samples were collected using a hand-held temperature probe and a Hydrosense II moisture probe (Campbell Scientific, Logan, Utah, USA), respectively. Measurements were also recorded continuously at different heights and depths at a meteorological tower (a single location) using a CS655 water content reflectometer (Campbell Scientific, Logan, Utah, USA) (Fig. S4).

**2.6. Data analysis**

All data were inspected, cleaned, transformed, statistically analysed and visualised using R software (R Core Team, 2022).

Data from the in-situ field experiment were visualised following the BACI principle (before-after-control-impact). The aim was to more efficiently distinguish between environmental versus treatment effects (Christie et al., 2019; Underwood, 1992). A two-sample Welch t-test was used to statistically test for differences in $N_2O$ fluxes before and after the start of the in-situ experiment. This was implemented via the t.test() function from the stats package in R. A Welch t-test was preferred over Student t-test as it provides more flexibility in case the variances of the different groups were not equal. Data on in-situ $N_2O$ fluxes were split into subsets according to the observed meteorological conditions (air temperature and soil moisture). Conditions where air temperature was above or below 12 °C were labelled as "warm" and "cold", respectively. Conditions where soil moisture was above or below 20% VWC were considered to be "wet" or "dry", respectively.

The laboratory N$_2$O flux data were subset to include only the first seven days of measurements in order to balance the samples collected during the first and second application periods. Welch Two Sample t-tests were also performed upon the laboratory cumulative fluxes of N$_2$O and NO (n = 3 per treatment) to test for any significant effect of C availability. No outliers have been removed from any of the datasets. Upon further examination, these were considered to represent genuine variation in N$_2$O fluxes rather than measurement error.

## 3. Results

### 3.1. Experimental NH$_3$ release

Between 1,000 and 10,000 standard litres of anhydrous NH$_3$ were released each month from September 2021 onwards, depending on wind conditions. Monthly NH$_3$ concentrations measured next to each static flux chamber at 0.5 m above the ground, increased from a maximum of 3.2 µg NH$_3$ m$^{-3}$ before the start of the experiment (July and August 2021) to a maximum

of 146.7 µg NH$_3$ m$^{-3}$ in response to NH$_3$ addition (Fig. 1). Experimentally increased concentrations of NH$_3$ were highest next to the chambers that were closest to the release line (median of 64.3 µg NH$_3$ m$^{-3}$ at a distance of 0.9 m) and decreased to nearly background concentrations beyond 30 m away from the source (median concentrations <3 µg NH$_3$ m$^{-3}$). Concentrations of NH$_3$ next to the control chambers were close to background levels (median range of 0.4 to 2.9 µg NH$_3$ m$^{-3}$). During the experiment, the enhanced atmospheric NH$_3$ concentrations resulted in a deposition gradient from approximately 3.6 to 71 kg

N ha$^{-1}$ yr$^{-1}$ and 12 to 162 kg N ha$^{-1}$ yr$^{-1}$ for soil surface and total deposition to all canopy layers, respectively (Deshpande et al., 2024).

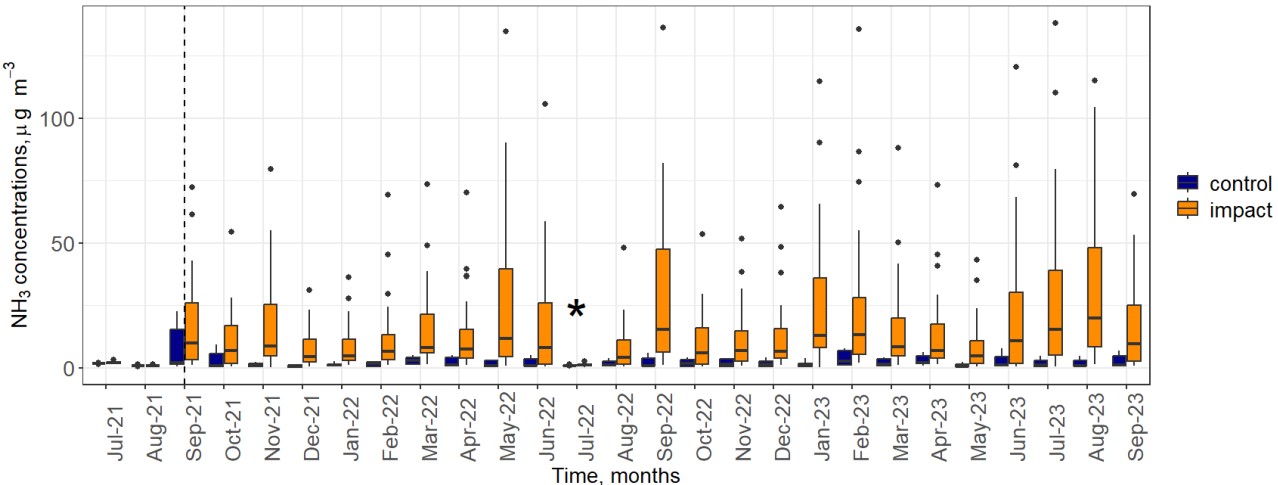

**Figure 1 Concentrations of NH$_3$ over the course of the experiment measured by ALPHA® samplers located at 0.5 m above the**
**ground and next to each static flux chamber (n = 36). The vertical dashed line denotes the time the NH$_3$ release system became active.**

**'Control' (blue) refers to chambers up wind of the NH₃ release system; 'impact' (orange) refers to chambers down wind of the system. An asterisk (*) in July 2022 indicates that the NH₃ release system was inactive for most of the month due to technical issues.**

### 3.2 In-situ flux measurements

Soil fluxes of $N_2O$ measured directly from the field site were generally low (maximum value of an individual flux was 1.2 nmol $N_2O$ m$^{-2}$ s$^{-1}$) and 57% of flux measurements were lower than the minimum detectable flux (0.05 nmol $N_2O$ m$^{-2}$ s$^{-1}$). Fluxes of $N_2O$ were higher before the start of NH₃ release (March to August 2021) compared to after (Fig. 2). Individual chamber fluxes of $N_2O$ after the start of NH₃ enhancement were below 0.6 nmol $N_2O$ m$^{-2}$ s$^{-1}$ (except for a single data point at 1.2 nmol $N_2O$ m$^{-2}$ s$^{-1}$), and there was little difference between chambers located in control or in impact areas. In contrast, individual fluxes of $N_2O$ before the NH₃ enhancement ranged from 0.04 to 1.2 nmol $N_2O$ m$^{-2}$ s$^{-1}$. The median $N_2O$ flux values for control and impact chambers were similar during the same month in most cases. Median $N_2O$ "after release" fluxes were higher in control compared to impact chambers in February 2022. Median "after release" fluxes were approximately equal during October 2021, April 2022, July 2022 and September 2022. In all other cases, fluxes from impact chambers were slightly higher compared to the control chambers. Some of the highest fluxes of $N_2O$ were observed in July and August 2021 (before the start of the NH₃ enhancement).

The majority of $CH_4$ fluxes were negative, both before and after the start of the experiment (Fig. S5), thus indicating potential uptake of $CH_4$. Fluxes of $CH_4$ from control and impact chambers were typically within less than 0.2 nmol m$^{-2}$ s$^{-1}$ of each other. Apparent uptake of $CH_4$ was somewhat higher after the start of NH₃ enhancement, especially during July, August and September 2022. However, the majority of $CH_4$ fluxes (78%) fell below the analytical limit of detection (0.58 nmol m$^{-2}$ s$^{-1}$) and hence no further statistical analysis was performed.

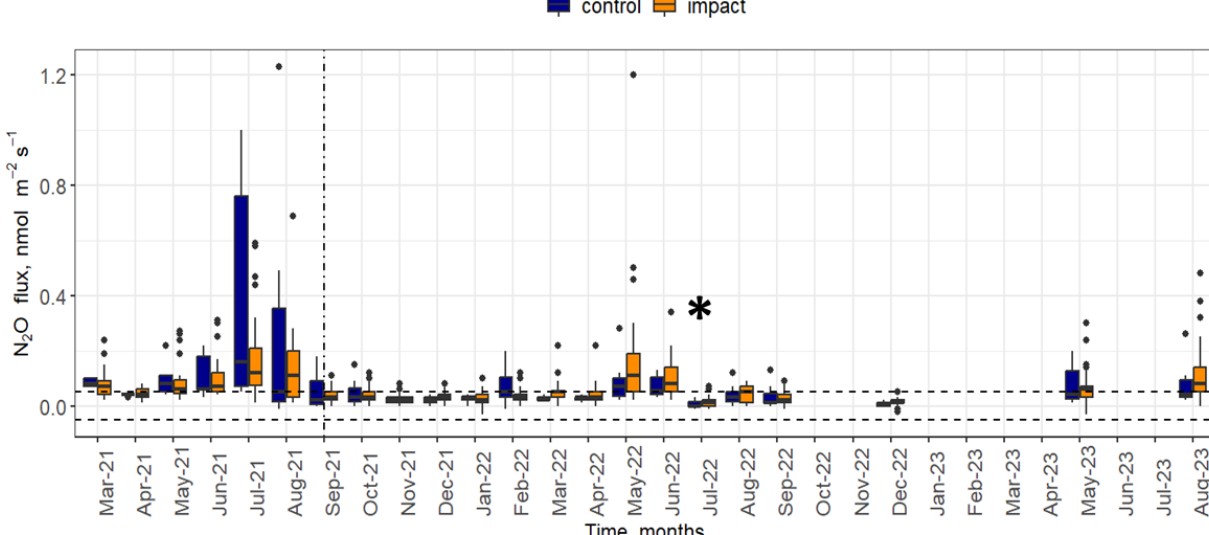

**Figure 2 Fluxes of N₂O over the duration of the experiment. Fluxes measured from control area (blue) and area where the impact of NH₃ deposition was expected (orange) are shown. The horizontal dashed lines mark the minimum detectable flux. The vertical line denotes the start of the NH₃ release. An asterisk (*) in July 2022 indicates that the NH₃ release system was inactive for most of the month due to technical issues.**

There was no clear relationship between NH₃ dry deposition and N₂O fluxes (linear regression, $R^2 = 0.1$, Fig. 3). Fluxes of N₂O at high levels of N deposition (>100 kg N ha⁻¹ yr⁻¹) were relatively low (<0.4 nmol N₂O m⁻² s⁻¹), and similar in magnitude to fluxes measured from the control area. There was a single data point which represented a relatively high N₂O flux (1.2 nmol N₂O m⁻² s⁻¹) at high N deposition (approximately 45 kg N ha⁻¹ yr⁻¹), however this was due to the commonly found lognormal distribution of natural N₂O fluxes rather than a treatment effect.

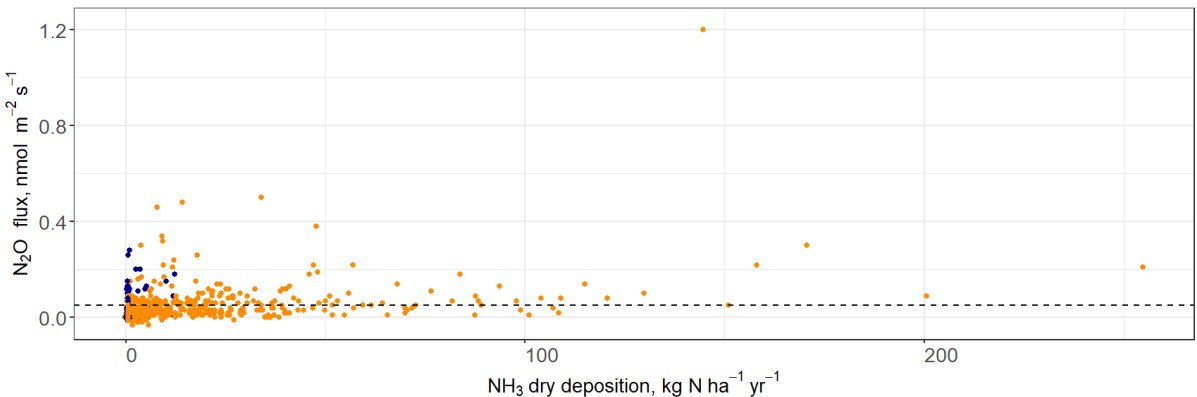

**Figure 3 A scatter plot which represents the relationship between NH₃ dry deposition and N₂O fluxes. Horizontal dashed line marks the minimum detectable flux. Fluxes from control and impact areas are shown in blue and orange, respectively.**

### 3.3. In-situ soil N availability

Soil inorganic N availability in the form of $NO_3$-N and $NH_4$-N changed little in response to the experimental $NH_3$ addition. The majority of soil $NO_3$-N concentrations (87%) were below 3 µg $NO_3$-N $g^{-1}$ (Fig. 4). There was little difference between inorganic N content in soils measured before and after the start of the experiment. The median values and the spread of the distribution of control and impact chambers were comparable during most months. One exception was March 2023 where median impact chamber $NO_3$-N concentrations were higher than the control chambers, though the spread of these values was large indicating high spatial variability.

Concentrations of $NH_4$-N were higher compared to $NO_3$-N (monthly medians ranged from 0.3 to 13.5 $NH_4$-N µg $g^{-1}$ and from 0.2 to 2.2 $NO_3$-N µg $g^{-1}$, for $NH_4^+$ and $NO_3^-$, respectively) and fluctuated more over the course of the experiment (Fig. 4). The highest median concentrations of $NH_4$-N were observed in April 2021, August 2022 and March 2023 (9.4, 10.0 and 13.5 µg $g^{-1}$, respectively). Median $NH_4$-N concentrations were higher for the control chambers compared to the impact chambers during all sampling months except for October 2021 and November 2022 (after the start of the experimental $NH_3$ addition).

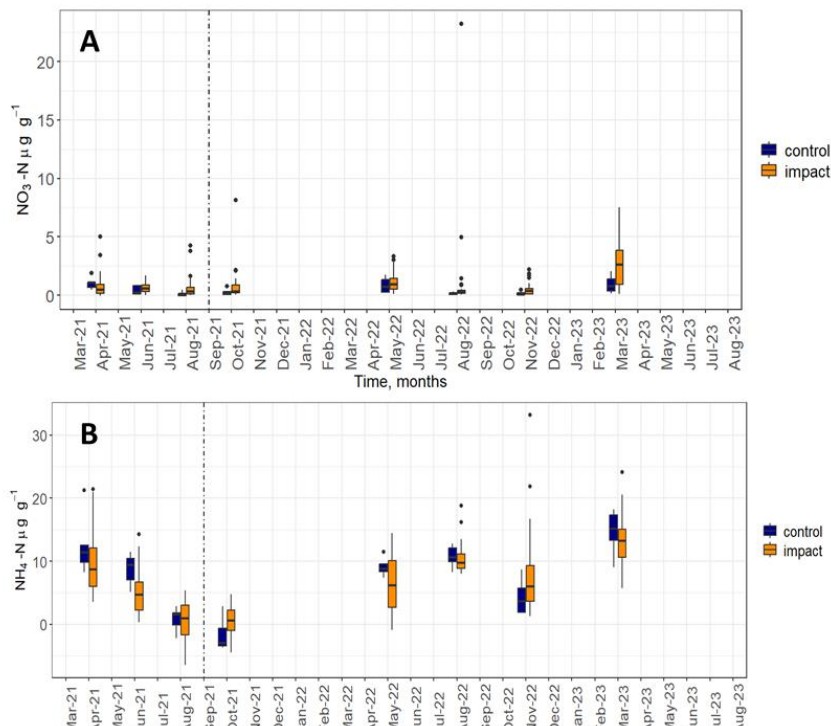

**Figure 4 Soil $NO_3$-N (A) and $NH_4$-N (B) concentrations in control (blue) and impact (orange) chambers over the duration of the experiment.**

### 3.4. In-situ soil temperature and soil moisture

Soil and air temperature as recorded by the meteorological station on site ranged between 2 and 16 °C and approximately -7 and 28 °C, respectively, and followed a seasonal pattern (Fig. S4). Values were consistent between the two years of the experiment.

Soil moisture also followed a seasonal pattern, although weekly variability was more pronounced compared to temperature
(Fig. S4). There was a discernible temporal trend in soil moisture, whereby July and August 2022 were particularly dry (VWC <13%). Soil moisture was generally low throughout the entire study period (<45% VWC).

Soil temperature and soil moisture combined had a pronounced effect on $N_2O$ fluxes (Fig. 5). Fluxes of $N_2O$ were the highest when the environmental conditions were simultaneously "warm" and "wet" (median $N_2O$ flux = 0.3 nmol m$^{-2}$ s$^{-1}$), which in
this study corresponded to soil temperatures >12 °C and VWC >20%. There was no difference among $N_2O$ fluxes when temperatures were <12 °C or when moisture was <20% VWC (median $N_2O$ flux = 0.05 nmol m$^{-2}$ s$^{-1}$). This indicates a dual threshold which is likely determining $N_2O$ fluxes.

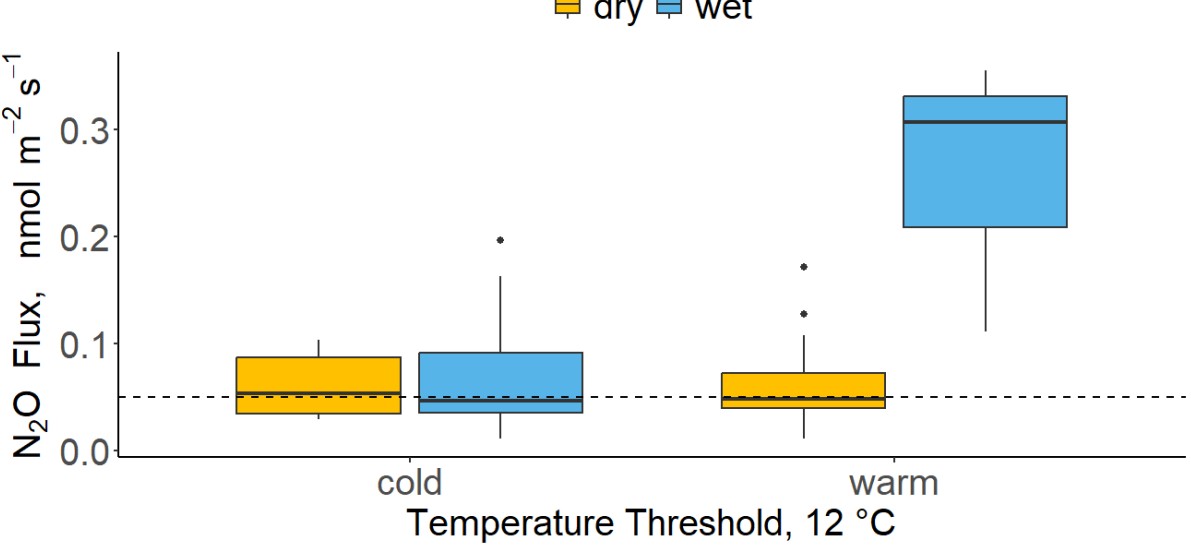

**Figure 5 A boxplot presenting daily average $N_2O$ fluxes at a dual threshold between soil temperature and soil moisture. Temperatures <12 °C were labelled as "cold" and >12 °C as "warm". The moisture threshold was set to 20% VWC, whereby values below the threshold were labelled as "dry" (orange) and the rest were labelled as "wet" (blue). Horizontal dashed line represented the minimum detectable flux. Soil temperature and moisture were recorded next to each chamber.**

### 3.5 Ex-situ impacts of soil carbon availability

The effects of C availability on $N_2O$ fluxes were tested during a laboratory incubation experiment, where air temperature and soil moisture were kept constant at 19 °C and approximately 30% VWC. Similarly to the in-situ field experiment, $N_2O$ fluxes did not increase in response to N addition under controlled laboratory conditions ($N_2O$ fluxes <2 ng $N_2O$-N $g^{-1}$ $d^{-1}$). This was the case for both reduced ($NH_4^+$) and oxidised ($NH_4NO_3$) forms of N at both medium (40 N) and high (100 N) levels of N addition and was consistent among the triplicate cores (Fig. 6). The highest $N_2O$ flux across all cores was measured following

the application of a source of labile C (glucose) alongside N. It occurred on day 37 of the experiment and ranged between 1317 ng $N_2O$-N $g^{-1}$ $d^{-1}$ (NH4 40N treatment) and 1801 ng $N_2O$-N $g^{-1}$ $d^{-1}$ (AN 40N treatment) across all cores, including the 0N controls (Fig. 6). The magnitude (1531 ng $N_2O$-N $g^{-1}$ $d^{-1}$) and duration of the peak (3 days) of the control (0N) were not significantly different from the experimental ones (t-test, p value >0.5).

Cumulative $N_2O$ fluxes were two orders of magnitude greater following the application of C and N together relative to the application of N only (t-test, p-value <0.01). These ranged from 13 (0N) to 71 ng $g^{-1}$ $N_2O$-N (AN 100N) and from 2416 (NH4 40N + C) to 3057 ng $g^{-1}$ $N_2O$-N (AN 40N + C) for the N and N+C applications, respectively. The mean cumulative emissions of application of 'N only' and 'N+C' were 2.5 ± 0.7 and 3091 ± 874 ng $N_2O$-N $g^{-1}$, respectively. Cores treated with AN exhibited higher cumulative $N_2O$ flux relative to cores treated with $NH_4^+$ during both application periods, although this

difference was not significant. In terms of N dose, 100N resulted in higher fluxes compared to 40N for all treatments except AN plus glucose.

    The primary pathway for N gaseous losses following the application of N only was NO rather than $N_2O$ fluxes, although their magnitude was relatively low (ranging from 1.2 to 13.8 ng NO-N $g^{-1}$ $d^{-1}$). Uncertainty around NO fluxes was higher compared

to $N_2O$ fluxes, likely due to a higher variability among the triplicates (Fig. S6). Fluxes of NO were low (<1.2 ng NO-N $g^{-1}$ $d^{-1}$) following the application of glucose irrespective of the treatment. Overall, gaseous N losses occurred predominantly in the form of $N_2O$ rather than NO during the laboratory experiment (two orders of magnitude higher cumulative fluxes) due to the effect of the glucose addition (Table 2).

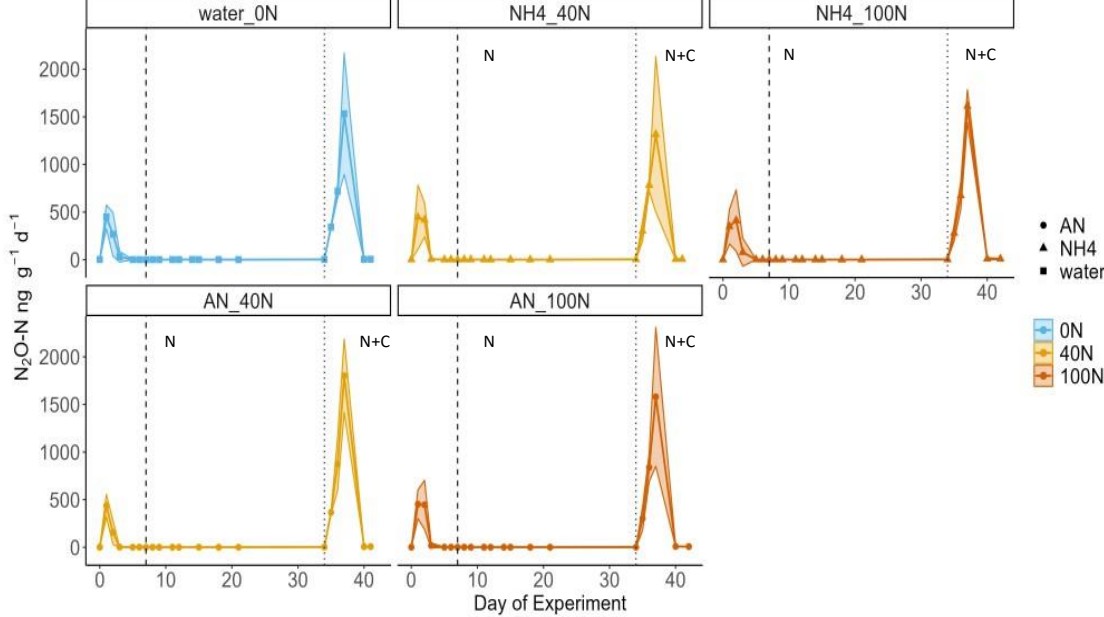

**Figure 6 Fluxes of N₂O over the course of the laboratory experiment. Vertical dashed lines correspond to the first N application (in the form of $NH_4^+$ or AN, depending on the treatment); vertical dotted lines signify the second application of N and glucose (N+C). Blue, orange and red correspond to N addition of 0, 40 and 100 kg N ha⁻¹ yr⁻¹, respectively. Shaded areas correspond to the 95 confidence intervals (C.I.) as calculated based on triplicates. The initial increase in N₂O fluxes following rewetting on day 1 was attributed to the well-documented Birch effect (Birch, 1964) and is not discussed further. Fluxes of N₂O-N are presented per grams of dry soil rather than per area to better represent the laboratory nature of the experiment.**

**Table 2 Cumulative fluxes of N₂O and NO at three levels of N addition (the equivalent of 0, 40 and 100 kg N ha⁻¹ yr⁻¹). Nitrogen was applied either in the form of ammonium ($NH_4^+$) or ammonium nitrate (AN, $NH_4NO_3$). NDF stands for "non-detectable flux".**

| N form | Treatment | Time (days) | Cumulative Flux, (N₂O-N, ng g⁻¹) | Cumulative Flux, (NO-N, ng g⁻¹) |
|---|---|---|---|---|
| Control (water) | 0N | 34 | 13 | 65 |
| $NH_4^+$ | 40N | 34 | 32 | 129 |
| $NH_4^+$ | 100N | 34 | 45 | 190 |
| AN | 40N | 34 | 60 | 279 |
| AN | 100N | 34 | 71 | 417 |
| Control (water) | 0N + C | 42 | 2603 | NDF |
| $NH_4^+$ | 40N + C | 42 | 2416 | NDF |
| $NH_4^+$ | 100N + C | 42 | 2589 | NDF |
| AN | 40N + C | 42 | 3057 | NDF |
| AN | 100N + C | 42 | 2747 | NDF |

## 4. Discussion

### 4.1 Experimental NH$_3$ release

Experimentally increased atmospheric concentrations of NH$_3$ exhibited a comparable pattern to the ones observed in the Whim bog N addition experiment, where a similar NH$_3$ enhancement system was utilised (Leeson et al., 2017; Leith et al., 2005). Background monthly atmospheric NH$_3$ concentrations at the Glencorse field site ($<3.2$ µg NH$_3$ m$^{-3}$) were of similar magnitude as previously reported for non-agricultural areas across the UK and Europe (Sutton et al., 2011). Background dry deposition at the site was estimated to be approximately 0.7 kg N ha$^{-1}$ yr$^{-1}$ before the start of the experiment (Deshpande et al., 2024), which fell below the critical loads for the UK which are currently considered to be around 10 and 12 kg N ha$^{-1}$ yr$^{-1}$ (Vanguelova et al., 2024).

The experimental deposition values are high compared to naturally observed deposition in European forests (Flechard et al., 2020), however, they are representative of forests located close to point sources of NH$_3$, such as chicken farms (Pitcairn et al., 2002, 1998). Deposition values were also consistent with previous N manipulation studies in forests that often simulate N deposition levels of 100 kg N ha$^{-1}$ yr$^{-1}$ or higher (Du et al., 2024; Liu and Greaver, 2009). The concentrations of total N in a commonly used bioindicator of N, moss tissue (Pitcairn et al., 2003; Salemaa et al., 2020), were also higher in the impact relative to the control areas (data not presented), thus suggesting that the experimentally added N deposited within the field site. It can therefore be concluded that the NH$_3$ release system was working as expected resulting in elevated NH$_3$ concentrations at the Glencorse site.

### 4.2 Environmental factors affecting N$_2$O flux

Median N$_2$O fluxes were generally lower and with smaller variability after the start of the NH$_3$ enhancement experiment compared to before, both for impact and for control chambers, regardless of the distance to the NH$_3$ source. This went against the expectations that N$_2$O fluxes would increase at higher levels of N deposition (Davidson et al., 2000; Deng et al., 2020). The classic Hole in the Pipe model proposed that N$_2$O fluxes would be higher at increased levels of N deposition as N acts as a substrate for the microbial processes of nitrification and denitrification (Firestone and Davidson, 1989). This has been supported by some experimental studies. For instance, in a meta-analysis of 33 studies from non-agricultural soils, Aronson and Allison (2012) reported that N-amended plots released more N$_2$O relative to control plots. However, the measured N$_2$O response to N addition weakened over the 23 years covered by the meta-analysis. Bühlmann et al. (2015) and Horváth et al. (2006) also proposed that elevated atmospheric N deposition induced higher N$_2$O fluxes from Swiss and Hungarian forest soils, respectively.

The environmental factors that drive N$_2$O production and emission from soils are complex (Butterbach-Bahl et al., 2013) and increased N availability does not always enhance N$_2$O fluxes. Liu and Greaver (2009) reviewed global N$_2$O emissions from

agricultural and natural systems and how the magnitude and direction of fluxes are influenced by ecosystem type, N form and level of N addition. They reported a 215% increase in $N_2O$ fluxes in response to N enrichment from agricultural systems. However, there was no clear dose-response in non-agricultural soils. Similarly, in a large study of 31 European forests, Flechard et al. (2020) found that $N_2O$ fluxes were not affected by levels of N deposition.

The small change observed in soil inorganic N availability in our experiments is a potential explanation for why $N_2O$ fluxes did not change with increased $NH_3$ deposition in our study. While atmospheric N deposition is a commonly used proxy for predicting $N_2O$ fluxes from soils (Hergoualc'h et al., 2019; IPCC, 2019, 2006), it does not always correlate with soil N availability - which can be a major driver of $N_2O$ production (Niu et al., 2016). This is consistent with the present findings, whereby inorganic N concentrations were not higher closer to the N source. Redding et al. (2016) suggested that it is soil N availability rather than atmospheric $NH_3$ concentrations that control $N_2O$ fluxes from forest soils. They demonstrated this empirically by exposing clay and sandy soils to low levels of N deposition (the equivalent of 30 kg N ha$^{-1}$ yr$^{-1}$). Their findings proposed 70 mg N kg$^{-1}$ soil as a threshold below which N deposition did not induce $N_2O$ flux. The concentrations of inorganic N ($NH_4$-N and $NO_3$-N) that were measured in the current study were generally low (<35 µg N g$^{-1}$ or the equivalent of <35 mg N kg$^{-1}$) and so fell under this proposed threshold.

The fact that $N_2O$ fluxes did not change in response to increased $NH_3$ concentrations (nor deposition) could be a result of $N_2O$ production and emission being controlled by a wide range of environmental factors, other than N availability (Baggs, 2008; Butterbach-Bahl et al., 2013). Notably, soil moisture, temperature, and C availability. This is consistent with the findings of this study which suggest that environmental factors had a more pronounced effect on $N_2O$ fluxes relative to the experimental treatment (gaseous $NH_3$ addition). For instance, the highest in-situ $N_2O$ fluxes were observed between June and August 2021 (Fig. 2) which corresponded to a relatively warm (soil temperature > 12 °C) and wet period (VWC > 20 %) (Fig. S4). In contrast, $N_2O$ fluxes during June to August 2022 were relatively low (median flux < 0.05 nmol $N_2O$ m$^{-2}$ s$^{-1}$), which could potentially be explained by a period of drought that summer (VWC < 15%). These findings suggest a dual temperature-moisture threshold which could be controlling soil $N_2O$ fluxes.

Soil temperature is generally considered to have an amplifying effect on $N_2O$ fluxes (Braker et al., 2010). This is supported by the current observations where $N_2O$ fluxes increased with temperature (especially above 12 °C) and were generally lower over the winter (non-growing) season. In addition, soil moisture has also been documented to modulate $N_2O$, whereby fluxes typically occur between 30% and 90% WFPS, with a peak around 60% WFPS (Davidson et al., 2000). It is likely that the moisture levels at Glencorse during this experiment were generally too low for denitrification to occur as it is a well-drained sandy mineral soil. Even though $N_2O$ can be produced through other microbial processes (such as nitrification), denitrification is considered to be a major pathway and it tends to occur in wetter soils, under anaerobic conditions (Butterbach-Bahl et al., 2013).

Carbon availability is another environmental factor which has been proposed to strongly influence $N_2O$ production from forest soils (Chapin et al., 2011). Soil C acts as a major source of energy for microorganisms and as such plays an important role in modulating soil microbial processes (Fontaine et al., 2003). Glencorse is a young forest (<50 years old) which could explain the relatively low total C and C:N ratio that was observed (Luyssaert et al., 2008). For instance, the C:N ratio at Glencorse soils was 13.1 on average. Previously reported values for C:N ratio in the topsoil of European deciduous broadleaf forests ranged from approximately 10 to 20 (Flechard et al., 2020) and between 17.5 and 20.0 for European birch forest (Cools et al., 2014). In another study, Cleveland and Liptzin (2007) proposed that the stoichiometric C:N:P ratio is consistent in soils (similar to the "Redfield ratio" in oceans) and is approximately 186:13:1, which corresponds to a C:N ratio of 14.3. The C:N ratio in this study was lower than these stoichiometric values and hence it is possible that there was soil carbon limitation at the site. This hypothesis was further supported by the complementary laboratory experiment.

Under controlled laboratory conditions, a peak of $N_2O$ appeared three days after applying N together with glucose to the soil but not when N was added alone. Furthermore, control cores, which did not receive any additional N, exhibited the same pattern after the application of glucose. Nitrogen form and dose did not have any significant effect on the peak, magnitude and duration of $N_2O$ fluxes, whereby control cores were not significantly different from the experimental cores. This further highlighted that C limitation was more important than N availability in the study soils in terms of $N_2O$ fluxes. This observation is consistent with previous research, where the role of soil carbon availability has been highlighted. For example, Weier et al. (1993) demonstrated that denitrification rates were low at high concentrations of N (100 kg N $ha^{-1}$) but in the absence of a labile source of C (denitrification rate <5 g N $ha^{-1}$ $d^{-1}$ at 60% WFPS) in a laboratory incubation experiment. In contrast, denitrification rates increased with increasing levels of C in the form of glucose (the equivalent of either 180 or 360 kg $ha^{-1}$) (maximum denitrification rate of 1309 and 2606 g N $ha^{-1}$ $d^{-1}$ for low and high levels of C, respectively). In another laboratory incubation study using temperate forest soils, Haohao et al. (2017) reported higher cumulative fluxes of $N_2O$ when N was added (either as $NH_4Cl$ or $KNO_3$) together with a source of C (glucose). This was further supported by a study of Scottish grassland soils where $N_2O$ fluxes were higher after the application of glucose, especially at lower soil moisture levels (Sanchez-Martín et al., 2008). These observations could be due to the fact that organic C acts as an electron donor during the anaerobic process of denitrification (Morley et al., 2014; Zumft, 1997). This is consistent with the present findings from the laboratory experiment whereby $N_2O$ fluxes were low (<2 ng $N_2O$-N $g^{-1}$ $d^{-1}$) before the application of C.

## 5. Conclusions

This study highlights carbon limitation as a major factor which modulates the impacts of elevated N levels on gaseous N fluxes in forest soils. Fluxes of $N_2O$ increased after applying a source of C but not in response to N addition alone. Another implication of this study is the potential of forests or tree belts to mitigate $NH_3$ pollution without incurring an $N_2O$ penalty, given the very

limited response of N$_2$O fluxes to elevated levels of N during both the field and laboratory experiments. However, impacts on sensitive forest species and ecosystems must be considered when designing NH$_3$ mitigation strategies, to ensure they remain below NH$_3$ critical levels and N critical loads.  In this study, it has been demonstrated that chronically adding small amounts of N, which is consistent with a nearby source of NH$_3$, did not increase N$_2$O from forest soils. However, the role of environmental factors, such as soil temperature, moisture and carbon availability had a more pronounced effect on N$_2$O fluxes in comparison to the experimental treatment (NH$_3$ deposition). This might suggest that N$_2$O emissions at the study site, and in similar temperate forests, are unresponsive to increased N$_r$, at least in the short term. However, the long-term impacts of N enhancement on N$_2$O fluxes remain unclear. The current observations are not inconsistent with the IPCC emission factor (EF) for non-agricultural systems of 1% (IPCC 2023) and a longer-term dataset from this experiment would be beneficial to better constrain an emission factor. The response of forest ecosystems to increased N levels might change with more prolonged exposure (Skiba et al., 1999). Overall, there is a need for more long-term field experiments on the effects of increased N availability on N$_2$O fluxes from natural and semi-natural ecosystems.

## Acknowledgements

The authors acknowledge United Kingdom Research and Innovation (UKRI) Global Challenge Research Fund (GCRF) for South Asian Nitrogen Hub (NE/S009019/1) for the financial support to this study. The Glencorse site is partially supported by NERC, through the UKCEH National Capability for UK Challenges Programme NE/Y006208/1.  We would like to thank Toby Roberts and Stella White for help with field work and laboratory analysis, Robert Nicoll and Neil Mullinger for their contribution towards setting up and maintaining the Glencorse field site, Netty van Dijk and Maude Grenier for vegetation surveys and collecting and processing moss samples, Joe Casillo and Gavin Sim for total C and N analysis, and Fred Duarte and Amy Stephens for ALPHA samplers analysis.

## Data availability

Data are currently undergoing preparation for submission to the Environmental Information Data Centre (EIDC; https://eidc.ac.uk/, EIDC, 2025).

## Competing interests

The contact author has declared that none of the authors has any competing interests.

## Author contributions

GYT – conceptualisation, data curation, formal analysis, investigation, methodology, visualisation, writing (original draft)

DR – conceptualisation, funding acquisition, methodology, resources, supervision, writing (review and editing)

MJ - conceptualisation, funding acquisition, methodology, resources, supervision, writing (review and editing)

AD – methodology, investigation, resources, visualisation, writing (review and editing)

NC – formal analysis, methodology, investigation, resources, visualisation, writing (review and editing)

PL – formal analysis, methodology, resources, software, writing (review and editing)

DH – methodology, investigation, resources, writing (review and editing)

AI - investigation, resources, writing (review and editing)

JD - conceptualisation, funding acquisition, methodology, resources, supervision, writing (review and editing)

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
