# Peer review of "Environmental conditions rather than nitrogen availability limit nitrous oxide $(N_2O)$ fluxes from a temperate birch forest"

_EGUsphere, 2025_

## Author Response (AR2)

**Reviewer 1**

I have thoroughly enjoyed reviewing this manuscript describing experiments to quantify impact of $NH_3$ deposition on $N_2O$ emissions from forest soil, and additionally impact of potential carbon limitation on the magnitude of emissions. The manuscript is well written, using clear and precise language, and contains sufficient detail to easily follow described work. In particular, methodology section is well developed and would be useful to early career researchers seeking to improve their understanding of these methodologies.

The topic itself is of importance due to large uncertainty associated with $N_2O$ emissions from non-agricultural soils and in response to $NH_3$ deposition. While tree belts are suggested as a potential measure to mitigate impacts of high $NH_3$ producing industries (i.e. pig and poultry farms) by capturing $NH_3$ plume, little is known of the impact on $N_2O$. This manuscript provides experimental data to address this knowledge gap and clearly shows that environmental conditions (soil moisture and temperature) as well as carbon availability can be the main drivers behind $N_2O$ emissions rather than N availability.

I only have few minor comments below:

1. L167, please correct information around analytical uncertainties-one number for $N_2O$ and one for $CH_4$

Thank you for highlighting a potential misinterpretation around the analytical uncertainty of $N_2O$ and $CH_4$. I have clarified this in text. Please see below.

"The analytical uncertainty in flux methodology was calculated to be $\pm 0.05$ nmol m$^{-2}$ s$^{-1}$ for $N_2O$ fluxes and $\pm 0.58$ nmol m$^{-2}$ s$^{-1}$ for $CH_4$ fluxes (Cowan et al., 2025)."

2. Some of the highest observed $N_2O$ fluxes occurred in July and August 2021, prior to the start of $NH_3$ release. I cannot find any narrative or explanation for this in the discussion section. Do you have any idea why this happened? What were the environmental conditions on the experimental site during this time?

Yes indeed, the period of the highest observed $N_2O$ fluxes coincided with a period of warm and wet conditions, which supports the dual temperature – moisture threshold suggested by this study. I have added this in lines 461 to 466.

"This is consistent with the findings of this study which suggest that environmental factors had a more pronounced effect on $N_2O$ fluxes relative to the experimental treatment (gaseous $NH_3$ addition). For instance, the highest in-situ $N_2O$ fluxes were observed between June and August 2021 (Fig. 2) which corresponded to a relatively warm (soil temperature > 12 °C) and wet period (VWC > 20 %) (Fig. S4). In contrast, $N_2O$ fluxes during June to August 2022 were relatively low (median flux < 0.05 nmol $N_2O$ m$^{-2}$ s$^{-1}$), which could potentially be explained by a period of drought that

summer (VWC < 15%). These findings suggest a dual temperature-moisture threshold which could be controlling soil $N_2O$ fluxes."

3. In L507-508 you say that observations in this study are consistent with the IPCC EF for non-agricultural soils. However, there is no mention anywhere in text what an indicative EF from the current experiment could be. Could you stipulate based on data from the control and impact chambers?

I have calculated $N_2O$ fluxes as a proportion of $NH_3$ dry deposition, which corresponded to 0.38% from control areas and 0.05% from impact areas (please see Table 1 below). According to the IPCC methodology fluxes from control areas are subtracted from fluxes from impact areas. In this case, this would result in a negative value. Even though fluxes of $N_2O$ were low throughout the study period (both in-situ and ex-situ), there was no evidence of uptake of $N_2O$ by the soil and previous work by Cowan et al.(2014) have suggested that negative $N_2O$ fluxes are often a methodological artefact. Presenting a negative emission factor could be misleading to the reader, which is why we have been reluctant to include these calculations in the manuscript.

Table 1 Fluxes of $N_2O$ as a proportion of $NH_3$ dry deposition.

| Chamber type | Mean $NH_3$ deposition, kg N $ha^{-1}$ $yr^{-1}$ | Mean $N_2O$ flux, nmol $m^{-2}$ $s^{-1}$ | $N_2O$ flux, kg N $ha^{-1}$ $yr^{-1}$ | $N_2O$ as a % from N deposition |
|---|---|---|---|---|
| control | 2.55 | 0.048 | 0.009 | 0.38 |
| impact | 20.55 | 0.054 | 0.011 | 0.05 |

**Reviewer 2**

This study assesses the effect of $NH_3$ deposition on $N_2O$ emissions in a temperate birch forest. $NH_3$ deposition was simulated in the forest for 2 years throuhg an $NH_3$ release system, and $N_2O$ emissions subsequently measured. Additionally, they performed laboratory studies with soils from the same forest, where they simulated N deposition under the different amounts of C. The study shows that $NH_3$ deposition did not lead to an increase in $N_2O$ emissions in the forest, which only occurred in the ex-situ/laboratory experiemnts in the presence of labile C and under warm and moist conditions. This shows that forest may be able to mitigate $NH_3$ pollution, at least on the short term, without leading to $N_2O$ penalies.

The topic is relevant and very well fits the scope of the journal. The study is well designed, and very well presented. The methodoloy used is appropiate to address the question. The soil incubations performed on top of the field trial are an excellent approach to understand the mechanisms that justify the observed resesults. The figures

are clear and the article is very well written. Therefore, I only have very minor suggestions:

4. Please clarify in the methods why the addition of N In the lab is done with both NH4+ and NH4NO3, instead of opting to only apply NH4+, which better simulates the addition of $NH_3$ as it was done in the field. This comes clear in the results, in order to check the effects of both reduced and oxidized N forms, but not before.

I have added a clarification in the methods section (line 186) as to why N was added both in the form of $NH_4^+$ and $NH_4NO_3$.

"N was applied in the form of either $NH_4^+$ in aqueous solution (Cowan et al., 2024) or $NH_4NO_3$ in order to study the effects of reduced and oxidised N forms."

5. In Fig 2 it could be indicated that the $NH_3$ release system was not working in July 22, as the previous 2 months $N_2O$ fluxes were high, and higher in the impact areas.

I have edited Fig.2 and its caption by adding an asterisk to indicate the temporary issue with the $NH_3$ release system in July 2022.

[Figure]

Figure 2 Fluxes of $N_2O$ over the duration of the experiment. Fluxes measured from control area (blue) and area where the impact of $NH_3$ deposition was expected (orange) are shown. The horizontal dashed lines mark the minimum detectable flux. The vertical line denotes the start of the $NH_3$ release. An asterisk (*) in July 2022 indicates that the $NH_3$ release system was inactive for most of the month due to technical issues.

6. Is there a reason in the field for the higher fluxes, before the start of the application of the $NH_3$ treatment, in the control areas? Fluxes reached very high levels in June-July 2021, which did not happen again. Can meteorologiocal conditions (temperture or soil moisture) at that time help explaining these extremely large fluxes?

Yes indeed, the period of the highest observed $N_2O$ fluxes coincided with a period of warm and wet conditions, which supports the dual temperature – moisture threshold suggested by this study. I have added this in lines 461 to 466. I have also addressed this as part of Reviewer 1, comment #2.

"This is consistent with the findings of this study which suggest that environmental factors had a more pronounced effect on $N_2O$ fluxes relative to the experimental treatment (gaseous $NH_3$ addition). For instance, the highest in-situ $N_2O$ fluxes were observed between June and August 2021 (Fig. 2) which corresponded to a relatively warm (soil temperature > 12 °C) and wet period (VWC > 20 %) (Fig. S4). In contrast, $N_2O$ fluxes during June to August 2022 were relatively low (median flux < 0.05 nmol $N_2O$ $m^{-2}$ $s^{-1}$), which could potentially be explained by a period of drought that summer (VWC < 15%). These findings suggest a dual temperature-moisture threshold which could be controlling soil $N_2O$ fluxes."

7. L384-385 and Table 2: I do not think the abbreviation AN was explained before, please clarify that this stands for NH4NO3.

I have edited Table 2 by explaining the meaning of the AN abbreviation. Please see below.

"Nitrogen was applied either in the form of ammonium ($NH_4^+$) or ammonium nitrate (AN, $NH_4NO_3$)."